# A Study of Vertical Structures and Microphysical Characteristics of Different Convective Cloud–Precipitation Types Using Ka-Band Millimeter Wave Radar Measurements

**Jiafeng Zheng [1,*], Peiwen Zhang [1], Liping Liu [2], Yanxia Liu [1] and Yuzhang Che [1,3]**

[1]  Plateau Atmosphere and Environment Key Laboratory of Sichuan Province, School of Atmospheric Sciences, Chengdu University of Information Technology, Chengdu 610225, China
[2]  State Key Lab of Severe Weather, Chinese Academy of Meteorological Sciences, Beijing 100081, China
[3]  Department of Mechanical Engineering, Tokyo Institute of Technology, Ookayama, Meguro-ku, Tokyo 152-8550, Japan
*  Correspondence: zjf1988@cuit.edu.cn; Tel.: +86-028-8596-6389

**Abstract:** Millimeter wave cloud radar (MMCR) is one of the primary instruments employed to observe cloud–precipitation. With appropriate data processing, measurements of the Doppler spectra, spectral moments, and retrievals can be used to study the physical processes of cloud–precipitation. This study mainly analyzed the vertical structures and microphysical characteristics of different kinds of convective cloud–precipitation in South China during the pre-flood season using a vertical pointing Ka-band MMCR. Four kinds of convection, namely, multi-cell, isolated-cell, convective–stratiform mixed, and warm-cell convection, are discussed herein. The results show that the multi-cell and convective–stratiform mixed convections had similar vertical structures, and experienced nearly the same microphysical processes in terms of particle phase change, particle size distribution, hydrometeor growth, and breaking. A forward pattern was proposed to specifically characterize the vertical structure and provide radar spectra models reflecting the different microphysical and dynamic features and variations in different parts of the cloud body. Vertical air motion played key roles in the microphysical processes of the isolated- and warm-cell convections, and deeply affected the ground rainfall properties. Stronger, thicker, and slanted updrafts caused heavier showers with stronger rain rates and groups of larger raindrops. The microphysical parameters for the warm-cell cloud–precipitation were retrieved from the radar data and further compared with the ground-measured results from a disdrometer. The comparisons indicated that the radar retrievals were basically reliable; however, the radar signal weakening caused biases to some extent, especially for the particle number concentration. Note that the differences in sensitivity and detectable height of the two instruments also contributed to the compared deviation.

**Keywords:** Ka-band millimeter wave radar; cloud–precipitation; vertical structure; microphysical characteristic

---

## 1. Introduction

Cloud and precipitation always experience multiple changes during their life cycles in the atmosphere. A reliable detection of these changes at different time stages using remote sensing tools is crucial to a better understanding of the macro- and microscopic structures and physical processes of cloud–precipitation. Furthermore, abundant measurements can also be valuable for applications in the fields of numerical weather prediction, weather modification, and climate related research [1–3].

In recent years, millimeter wave cloud radar (MMCR) has been used as a primary tool to observe cloud–precipitation, which typically involves Ka and W bands with wavelengths of ~8 and ~3 mm, respectively. Considering wavelength and attenuation, the Ka band is more appropriate than the W band with which to observe common cloud and weak precipitation, while the W band is more suitable for fog or weak cloud observations. Compared to the commonly used centimeter wave radar, MMCR has several advantages for cloud measurement. First, MMCR can be more sensitive to small cloud particles and the received signal is more remarkable with a larger signal-to-noise ratio (SNR), because the backscattered ability of a hydrometeor is inversely proportional to the fourth power of an electromagnetic wavelength. Second, MMCR typically operates with a narrow radar beam width. As a result, the measurement is less affected by ground clutter, and the radar can obtain a high antenna gain [4]. Moreover, the narrow beam width can avoid some other non-hydrometeor signals (e.g., clear air turbulence), thereby leading to a good radar return that consists only of cloud and precipitation particles [5–7]. By transmitting short pulses, MMCR also has very high spatial and temporal resolution, with magnitudes of several decameters and a few seconds, respectively [8–10]. This kind of high spatiotemporal resolution is superb for multiple aspects of radar data application, ranging from cloud microphysical phenomena to dynamic processes.

MMCR has been designed and employed widely in different campaigns for the measurement and research of cloud–precipitation because of its remarkable advantages against other instruments. In 1996, the US Department of Energy launched the Atmospheric Radiation Measurement (ARM) project. The ARM project successively deployed Ka-band MMCRs in some climatological regions in Oklahoma, Alaska, Manus Island, Nauru Island, etc. After appropriate data processing, the Ka-band radar measurements were utilized to study the macro- and microscopic characteristics of different clouds above the ground sites, and the relationships between cloud radiation effects and climate change [11,12]. The ARM project also bridged the gap between ground-based observation and satellite remote sensing by operating these MMCRs to continuously monitor cloud-related variables over multiyear time periods [13,14]. In Europe, since 2001, several ground sites equipped with MMCRs have been established to conduct a long-term observational project named CloudNet. This project has deployed both Ka- and W-band MMCRs at three locations in different countries, namely, Cabauw (The Netherlands), Chilbolton (the United Kingdom), and Palaiseau (France). The MMCRs, combined with lidar, a ceilometer, and a microwave radiometer, can provide vertical profiles of cloud and aerosol properties at high temporal and spatial resolutions. The observational data were used to evaluate the representations of clouds in climate and weather forecast models, and a couple of studies have demonstrated that this kind of operational network has strong potential to improve the representation of clouds in numerical models [15,16]. In 2013, China began the third Tibetan Plateau Atmosphere Science Experiment. Two solid-state transmitter Ka-band MMCRs manufactured by the Chinese Academy of Meteorological Sciences and the China Aerospace Science & Industry Corporation were deployed in Nagqu and Nyingchi during the summers of 2014 and 2015, respectively. The MMCRs were equipped with a vertically pointing antenna, and simultaneously operated under four different working modes to meet the requirements of different cloud observation overheads to a large extent. The long-term, high-resolution radar products have been used mainly to reveal the diurnal cloud variation, vertical structure, and microphysical characteristics over the Tibetan Plateau [17–19]. As a significant advancement, MMCR has also been installed on satellites to monitor cloud and precipitation from space without topographic limitations. In 2006, the National Aeronautics and Space Administration (NASA) successfully launched the CloudSat satellite, which carried a W-band cloud profiling millimeter wave radar (CPR). The CPR is the first space-based MMCR, and can provide observations to advance our understanding of cloud abundance, distribution, structure, and radiative properties [20]. In addition, in 2013, NASA also launched the global precipitation measurement (GPM) satellite, which was equipped with a Ku/Ka dual-wavelength radar. As an upgraded alternate to the Tropical Rainfall Measuring Mission satellite, the GPM has a broader range for global cloud and precipitation monitoring, and can distinguish between different hydrometeor phases [21].

In a specific application of cloud and precipitation physics research, the high sensitivity and high spatiotemporal measurement of MMCR can be used to reveal fine vertical structure and physical property variation in cirrus, cumulus, stratus, stratocumulus, and other clouds. It helps promote our understanding of different physical processes and changes in various cloud and precipitation events during their life cycles [9,22–25]. By using some specific algorithms, MMCR measurements can also produce many useful macro and micro quantities about cloud and precipitation, such as cloud top height, cloud base height, cloud fraction, cloud type, optical thicknesses, particle size distribution, rain rate, and water content. These observational quantities are essential for improving parameterization schemes in the current cloud and weather simulation numerical models [26–30]. Focusing on convection, although convective rainfall will attenuate the MMCR, studies have shown that the radar measurement remains available and has the potential to characterize air motion, microphysical changes, hydrometeor phases, and their relationships in the interior of the convection [31,32]. Despite the beneficial usage of MMCR for convection observations, studies of convection in South China are still rare. Therefore, this manuscript focuses mainly on the vertical structure and microphysical characteristics of different types of convective cloud–precipitation in South China during the pre-flood season, using a vertical pointing Ka-band MMCR. The remainder of this study is arranged as follows: Section 2 presents a description of the instruments, data, data processing procedures, and cloud–precipitation microphysical parameter retrieval methods used herein; Section 3 gives the results, including the weather background, vertical structure, and microphysical property observational results of four different kinds of convective cloud–precipitation; Section 4 discusses the difference between radar retrieval and ground observation; and Section 5 ends the paper with a summary.

## 2. Materials and Methods

### 2.1. Instruments and Data

To promote understanding of cloud and precipitation during the pre-flood season in South China, the Chinese Academy of Meteorological Sciences conducted a field experiment in April 2016 in Longmen (23.783°N, 114.25°E, 86 m), Guangdong Province, China, using a suite of remote sensing instruments.

A Ka-band MMCR accompanied by a raindrop disdrometer was used to detect the cloud–precipitation profiles and the ground rain drop size distribution (DSD). The MMCR was designed with a solid-state transmitter and equipped with an independent container to ensure a long-term stable working capacity. The radar system operated at 33.44 GHz, with a wavelength of 8.9 mm and a peak power over 100 W. A 2 m diameter antenna was installed at a vertical incidence angle on the top to achieve a 53 dB gain and form a 0.3° beam width able to lead to a high horizontal resolution (only 26 m at 5 km). By transmitting 0.2 µs pulses, the radar-related vertical resolution was 30 m. To observe the cloud and precipitation at different heights with different intensities, the radar system had three different operational modes (i.e., boundary layer, common, and cirrus modes) designed by configuring with different signal processing parameters. These modes were periodically operated, thereby resulting in an approximately 9 s temporal resolution. With a high spatiotemporal resolution, this radar was able to provide a fine-resolution mapping of the cloud–precipitation structures and boundaries [33,34]. The MMCR has a high sensitivity and can detect a −38-dBZ target at 5 km. The backscattering cross-sections of this radar can be higher than the S-band radar by 42 dB, under the Rayleigh scattering condition.

The radar used herein measured the original Doppler spectra, reflectivity, mean Doppler velocity, spectrum width, linear depolarization ratio, etc. Table 1 summarizes its key technical specifications.

**Table 1.** The Ka-band millimeter wave cloud radar (MMCR) key technical specifications.

| Items | | Technical Specifications |
|---|---|---|
| Radar system | | Doppler, solid-state, depolarization |
| Frequency | | 33.44 GHz |
| Wavelength | | 8.9 mm |
| Transmitted peak power | | $\geq$100 W |
| Antenna gain | | 53 dB |
| Beam width | | 0.3 degree |
| Pulse width | | 0.2 µs, 12 µs |
| Pulse repetition frequency | | 8333 Hz |
| Gate number | | 510 |
| Sensitivity | | −38 dBZ at 5 km |
| Resolutions | Horizontal resolution | 26 m at 5 km |
| | Vertical resolution | 30 m |
| | Temporal resolution | ~9 s (adjustable) |
| Detectable range | Height | 150 m–15.3 km |
| | Measurable reflectivity range | −50–30 dBZ |
| | Unambiguous velocity range | −18.54–+18.54 m·s$^{-1}$ |
| Measurements | Original data | Doppler velocity spectra |
| | Spectral moments | Reflectivity, mean Doppler velocity, spectrum width, linear depolarization ratio |

The OTT particle size velocity (Parsivel) disdrometer (OTT Hydromet, Germany), employed widely to measure the raindrop spectra in recent decades [35–37], was deployed collocated with the MMCR at the Longmen site. Parsivel is a laser-optical disdrometer that can simultaneously detect raindrop size and falling speed based on the signal attenuation caused by the precipitation particles as they pass through the laser beam. The particle size and the falling speed are determined by the signal attenuation degree and the crossing time, respectively. The emitted laser wavelength was 785 nm, with a frequency of 50 kHz. The measure height was only 1.4 m above the ground, with a sampling area of 54 cm$^2$ and a sampling time of 60 s. The detectable range of the particle diameter was 0.062–24.5 mm. The observed data were eventually reserved by division into 32 non-equidistant channels. Furthermore, the integrated rain rate, reflectivity, number concentration, mean diameter, and etc. was also able to be derived from the Parsivel observations. Table 2 shows the main technical specifications of Parsivel.

**Table 2.** The OTT particle size velocity (Parsivel) disdrometer main technical specifications.

| Items | Technical Specifications |
|---|---|
| Sensor type | Laser |
| Frequency | 50 KHz |
| Operating power | $\geq$2 W |
| Sampling height | 1.4 m |
| Sampling area | 54 cm$^2$ |
| Measurable particle type | Solid, mixed, liquid |
| Measurable particle diameter range | 0.062–24.5 mm (32 channels) |
| Measurable particle falling speed | 0.05–20.8 m·s$^{-1}$ (32 channels) |
| Measurement | Particle size (32 channels), particle falling speed (32 channels), particle number, rain rate, reflectivity, accumulated rain amount |

### 2.2. Data Processing and Cloud–Precipitation Microphysical Parameter Retrieval

Before analysis, appropriate data processing techniques and retrieval methods needed to be used to acquire reliable radar spectral moments, ground rainfall quantities, and microphysical parameters (Figure 1). Each step is described in detail in the following subsections.

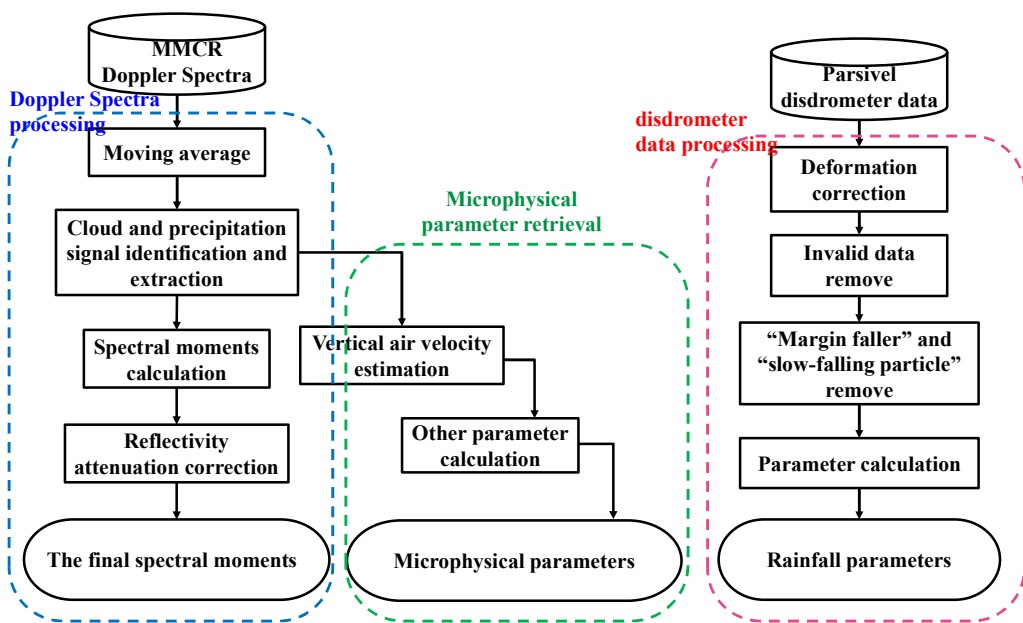

**Figure 1.** The flow chart of data processing and microphysical parameter retrieval for MMCR and Parsivel disdrometer.

### 2.2.1. Radar Doppler Spectra Processing

For a vertically pointing MMCR, Doppler spectra are the primary data, containing abundant information about the cloud microphysical and dynamic features. Thus, to obtain more accurate radar spectral moments and provide the necessary foundation for further retrieval, an appropriate spectra post-processing procedure was used.

First, a three-point moving average was performed on each spectrum comprising 256 bins to slow down the radar noise fluctuation to some extent. The noise level representing the mean power of the radar noise was estimated using a method proposed by Monique et al. [38].

Second, according to the noise level, all spectral bins can be preliminarily distinguished as a noise or a signal, that is, if the power of a spectral bin is larger than the noise level, it was regarded as a signal; otherwise, it was noise. The cloud–precipitation signals in the radar spectra generally had a specific width and SNR because of the different sizes and velocities of the particles in the radar sampling volume. A signal can specifically be an effective cloud–precipitation signal when the number of signal bins exceeds five, and the SNR value is higher than −12 dB [39]. Except for this condition, the signal was still treated as radar noise. The corresponding Doppler velocities of the start, end, and peak bins of the cloud–precipitation signals were also recorded for further use.

Third, after identifying the cloud–precipitation signals, the recorded Doppler velocity information was used to calculate the five radar spectral moments, namely, the total signal power (zeroth moment), mean Doppler velocity (first moment), spectrum width (second moment), spectral skewness (third moment), and spectral kurtosis (fourth moment). These five moments are closely related to the inner-cloud hydrometeor size, motion state, distribution, and phase. The total signal power $P_{total}$ (Equation (1)) represents the scattered energy of all the particles in the radar sampling volume. After calibration, $P_{total}$ transforms into $P'_{total}$, which is the total power directly received by the radar antenna that can be used to compute the reflectivity Z ($mm^6m^{-3}$) through the radar equation (Equation (2)). The mean Doppler velocity $\overline{V}$ ($m{\cdot}s^{-1}$) and the spectrum width $\sigma_v$ ($m{\cdot}s^{-1}$) can be calculated by Equations (3) and (4), respectively,

$$P_{total} = \sum_{V_s}^{V_e} (P_s - P_n) \tag{1}$$

$$Z = \frac{P'_{total} \times R^2}{C}, \ C = \frac{P_t \times G^2 \times \theta \times \varphi \times h \times \pi^3 \times |k|^2}{1024 \times \ln 2 \times \lambda^2 \times L_\varepsilon} \tag{2}$$

$$\overline{V} = \frac{\sum_{i=V_s}^{V_e} i \times (P_s - P_n)}{\sum_{i=V_s}^{V_e} (P_s - P_n)} \tag{3}$$

$$\sigma_v = \sqrt{\frac{\sum_{i=V_s}^{V_e} \left(i - \overline{V}\right)^2 \times (P_s - P_n)}{\sum_{i=V_s}^{V_e} (P_s - P_n)}} \tag{4}$$

where $V_s$, $V_e$ ($m{\cdot}s^{-1}$) are Doppler velocities of the start signal bin and the end signal bin; $P_s$ (dBm) is the signal power; $P_n$ (dBm) is the noise level; C represents the radar constant; R (km) is the distance from the radar to the target; $P_t$ (dBm) is the radar transmitted power; $\theta$ and $\varphi$ (degree) are the radar horizontal and vertical beam widths; h (m) represents the spatial pulse length; $\lambda$ (mm) for wavelength; $|k|^2$ is the complex refractive index; and $L_\varepsilon$ (dB) is the feeder loss.

The radar can receive the electromagnetic wave signal from both the parallel and cross-polarization returns; therefore, the linear depolarization ratio LDR (dB) can be written as

$$LDR = 10 \log_{10} \frac{Z_V}{Z_H} \tag{5}$$

where $Z_V$ and $Z_H$ are the parallel and cross-polar reflectivity, respectively. The LDR is closely related to the particle phase. Its value can be very large for a sizable, non-spherical ice crystal or a particular melting particle, but is much smaller for a small spherical particle.

Spectral skewness $S_k$ and kurtosis $K_t$ are two indicators used to describe the symmetry and flatness of the Doppler spectra. The radar Doppler spectra can generally approximately exhibit a Gaussian distribution in a stable cloud layer, with $S_k$ and $K_t$ values both close to zero. However, the values quickly become positive or negative if the cloud droplets have developed into raindrops or their phases have changed. Kollias et al. [40,41] pointed out that $S_k$ and $K_t$ are very sensitive to

the formation of a drizzle in the cloud, and can be good indicators for drizzle identification. The calculation of $S_k$ and $K_t$ is presented below:

$$S_k = \frac{\sum_{i=V_s}^{V_e} \left(i - \overline{V}\right)^3 \times (P_s - P_n)}{\sigma_v^3 \times \sum_{i=V_s}^{V_e} (P_s - P_n)} \tag{6}$$

$$K_t = \frac{\sum_{i=V_s}^{V_e} \left(i - \overline{V}\right)^3 \times (P_s - P_n)}{\sigma_v^4 \times \sum_{i=V_s}^{V_e} (P_s - P_n)} - 3 \tag{7}$$

As the Ka-MMCR electromagnetic wave penetrates cloud and precipitation, the returned radar signal can be gradually weakened because of attenuation by the hydrometeor. In this case, the radar measured Z will be underestimated to a certain extent. Thus, an iterative procedure was further used to correct the Z. The procedure was based on the following relationships [42],

$$k_i = \alpha Z_{correct}(i)^{\beta} \tag{8}$$

$$\tau_i = \tau_{i-1} \times \exp(-2 \times k_i \times \Delta R) \tag{9}$$

$$Z_{correct}(i) = \frac{Z(i)}{\tau_{i-1}} \times \exp(k_i \times \Delta R) \tag{10}$$

where i represents the gate number, k (dB·km$^{-1}$) is the attenuation coefficient, $\tau$ (dB) is the two-way transmissivity, $Z_{correct}$ is the radar reflectivity after attenuation correction, and $\Delta R$ is the radar gate length. Attenuation is a continuous process yielded from near to far in the radar radial, therefore, the $Z_{correct}$ will be calculated gate-to-gate. To start the iteration, the initial $\tau_0$ and $Z_{correct}(0)$ was set to 1 and Z(0), respectively. The coefficients $\alpha$ and $\beta$ were set to 0.00334 and 0.73, respectively [43].

### 2.2.2. Cloud–Precipitation Microphysical Parameter Retrieval

Some key microphysical parameters of cloud–precipitation, including air vertical velocity, mean particle diameter, total number concentration, liquid water content, and rain rate, were derived for further analysis based on the radar Doppler spectra post-processing. From a technical perspective, the vertical air velocity ($V_{air}$) is a prerequisite for the retrieval of other parameters. Several pioneer studies attempting to retrieve the $V_{air}$ were often based on fixing an empirical relationship between the radar Z and terminal velocity of hydrometeor ($V_t$), because both Z and $V_t$ are proportional to hydrometeor diameter [44,45]. However, a straightforward relationship among the variables is difficult to establish, and is not known in the case of convective clouds and precipitation. A relatively new technology called "small-particle-traced" was proposed by Gossard and Kollias and then applied by Shupe, Zheng, and Sokol in different clouds and precipitation [9,18,26,46,47]. This method is based on the high sensitivity of MMCR, which has sufficient ability to detect small particles. It regards the smallest particle, which has a negligible falling speed compared with the air motions in the convective cloud (typically, one or two orders of magnitude larger), as a tracer, and deduces $V_{air}$ using the velocity of the left-edge cloud signal in the Doppler spectrum. This approach can be more valid than the Z–$V_t$ empirical relationship because there are no preparatory assumptions [18]. However, undeniably, this method may suffer from some bias when the radar detects rainfall, in which the traced particles have a non-negligible terminal velocity. Zheng et al. preliminarily verified the retrieved $V_t$ in convective clouds and precipitation; they found that the retrieval result by this method was still quite reliable, despite suffering from some biases [18]. Considering this, we used this method to retrieve the $V_{air}$ in our study. In addition, in Section 4, a comparison shows that the radar retrieved mean diameter agreed well with reality. This indirectly indicates that the retrieval method was reliable.

Second, the Doppler spectra were shifted according to $V_{air}$, and the power of each cloud signal was converted from dBm to dBZ using Equations (1) and (2). The relationship between particle

terminal velocity and diameter must be determined before the retrieval of the remaining parameters. This relationship is very complicated for an ice hydrometeor, but can be quite simple and accurate for a liquid particle. The relationship for a liquid particle is presented as follows [48,49]:

$$D = \frac{1}{0.6} \times \ln\frac{10.3}{9.65 - V_t/\delta(h)} \tag{11}$$

$$\delta(h) = 1 + 3.68 \times 10^{-5} h + 1.71 \times 10^{-9} \, h^2 \tag{12}$$

where D (mm) represents the particle diameter; $\Delta D_i$ (mm) is the diameter interval; $V_t$ (m/s) is the terminal velocity of the particle; h (m) is the particle height (above sea level); and $\delta(h)$ is a correction factor. The particle size distribution $N(D_i)$ (m$^{-3}$), rain rate R (mm·h$^{-1}$), liquid water content LWC (g·m$^{-3}$), particle mean diameter $D_m$ (mm), and total number concentration $N_{total}$ (m$^{-3}$·mm$^{-1}$) can be directly calculated as follows [50]:

$$P_{D_i} = \frac{C \times D_i^6}{R^2} \tag{13}$$

$$N(D_i) = \frac{P_i}{P_{D_i} \times \Delta D_i} \tag{14}$$

$$R = \frac{\pi}{6} \sum_{i=D_{min}}^{D_{max}} D_i^3 \times V_t(D_i) \times N(D_i) \times \Delta D_i \tag{15}$$

$$LWC = \frac{\pi}{6} \sum_{i=D_{min}}^{D_{max}} \rho \times D_i^3 \times N(D_i) \times \Delta D_i \tag{16}$$

$$D_m = \frac{\sum_{i=D_{min}}^{D_{max}} D_i \times N(D_i) \times \Delta D_i}{\sum_{i=D_{min}}^{D_{max}} N(D_i) \times \Delta D_i} \tag{17}$$

$$N_{total} = \sum_{i=D_{min}}^{D_{max}} N(D_i) \times \Delta D_i \tag{18}$$

where $P_{D_i}$ (mW) is the theoretical power caused by a single particle with a diameter of $D_i$, $P_i$ (mW) is the radar measured power for the particles with a diameter of $D_i$, $D_{min}$ and $D_{max}$ (mm) represent the detected minimum diameter and maximum diameter in Doppler spectra, and $\rho$ (g·cm$^{-3}$) is the water density.

### 2.2.3. Parsivel Data Post-Processing

Existing studies [51–53] have pointed out that raindrop spectra observed by Parsivel need further post-processing to correct observational errors. Therefore, we further processed the Parsivel data following the procedure described below.

First, Parsivel detected the DSD based on the assumption that a falling particle is spherical because the laser can measure only in the horizontal direction. However, in nature, a raindrop is usually ellipsoidal, as its diameter is larger than 1 mm. Thus, correction for the measured diameter of raindrop was performed (Equation (19)) using the method proposed by Battaglia et al. [54]:

$$D = \begin{cases} D_{Par} & (D_{Par} \leq 1 \, \text{mm}) \\ (1.075 - 0.075 \times D_{Par}) \times D_{Par} & (1 \, \text{mm} < D_{Par} \leq 5 \, \text{mm}) \\ 0.7 \times D_{Par} & (D_{Par} > 5 \, \text{mm}) \end{cases} \tag{19}$$

where $D_{Par}$ (mm) is the original measured raindrop diameter and D (mm) represents the equivalent spherical diameter of the raindrop after correction.

Second, considering the actual sensitivity of Parsivel, the data in the first two channels corresponding to the smallest particles were abandoned [51,55]. Any data with diameters greater than 6 mm were also deleted, because of the fact that a raindrop will break up when its diameter exceeds 6 mm.

Third, we should pay more attention to two kinds of inaccurate data. One is called the "margin faller," which states that a raindrop might be misclassified as a small particle falling faster than the others observed at the same size when it partially passes the laser beam. The other one is the "slow-falling particle," which is a raindrop of a certain size with an unrealistic falling speed caused by the effects of wind shear or splashing from other large raindrops hitting the instrument surface during heavy rainfall. The raindrops outside ±60% of the empirical fall velocity–diameter relationship were eliminated from the observed dataset to largely avoid the impact caused by the mentioned data inaccuracy problems [51,56].

The DSD in size channel i, $N(D_i)$, were calculated as follows based on the data processing procedures above:

$$N(D_i) = \sum_{j=1}^{32} \frac{n_{ij}}{A \times \Delta t \times V_j \times \Delta D_i} \tag{20}$$

where $D_i$ (mm) is the volume-equivalent diameter for the channel i; $N(D_i)$ ($m^{-3} \cdot mm^{-1}$) is the number concentration of raindrop per unit volume with diameters in an interval from $D_i$ to $D_i + \Delta D_i$, $n_{ij}$ represents the raindrop number within size channel i and velocity channel j; A ($m^2$) is the sampling area; and $\Delta t$ (60 s) is the sampling time. For a later comparison with the radar observation, the reflectivity $Z_p$, raindrop mean diameter $D_{mp}$, rain rate $R_p$, and total number concentrate $N_{total\_p}$ were also computed using the following equations:

$$Z_p = \sum_{i=1}^{32} \sum_{j=1}^{32} D_i^6 \times \frac{n_{ij}}{A \times \Delta t \times V_j} \tag{21}$$

$$D_{mp} = \frac{\Sigma_{i=1}^{32} D_i^3 \times N(D_i) \times \Delta D_i}{\Sigma_{i=1}^{32} N(D_i) \times \Delta D_i} \tag{22}$$

$$R_p = \frac{6\pi}{10^4} \sum_{i=1}^{32} \sum_{j=1}^{32} D_i^3 \times \frac{n_{ij}}{A \times \Delta t} \tag{23}$$

$$N_{total\_p} = \sum_{i=1}^{32} N(D_i) \times \Delta D_i \tag{24}$$

## 3. Results

A representative weather process during the pre-flood season with different convective cloud–precipitations that occurred in Longmen, Guangdong Province, China, on 21–23 April 2016 is analyzed in detail herein to investigate the application of the MMCR and disdrometer measurements in the study of cloud structures and microphysical processes. This section mainly shows the detailed background information and the corresponding analysis results.

### 3.1. Weather Background and Convection Evolution

The Longmen site was continuously affected by the low-latitude short trough, jet stream, and southwest airflow around the periphery of the Pacific subtropical high during the period 21–23 April 2016. Figure 2 presents a group of weather charts at both 700 hPa and the surface on 21–23 April. At 08:00 BJT on 21 April, two short troughs appeared in the west and north of the local site (Figure 2a). The warm and wet southwest airflow at the front of the west trough moved eastward to the site

because of the influence of the subtropical high. Meanwhile, the north trough made the dry and cold airflow from a higher latitude veer to the east and meet the warm and wet airflow around the local site. Three short troughs appeared at 08:00 BJT on 22 April. The first was located just over the station. The second and third troughs were located in the northwest and southwest of the station, respectively. These troughs continually transported warm and humid ocean airflow to the local site and provided a favorable lifting condition. At 08:00 BJT on 23 April, although no trough existed, the low-altitude jet stream was located near the station to bring in abundant water vapor. The weather charts of the surface showed that different fronts continually originated from the higher latitude and moved from the north to the local site. On 21 April, a cold front moved southward and arrived near the local station at 08:00 BJT. On 22 April, a stationary front was located in the north of the station. On 23 April, another cold front appeared in the north of the station and moved gradually from the north to the south. In summary, controlled by the short trough, jet stream, subtropical high, and front activity, the Longmen site in the Guangdong area had sufficient water vapor and an appropriate lifting condition for cloud–precipitation formation.

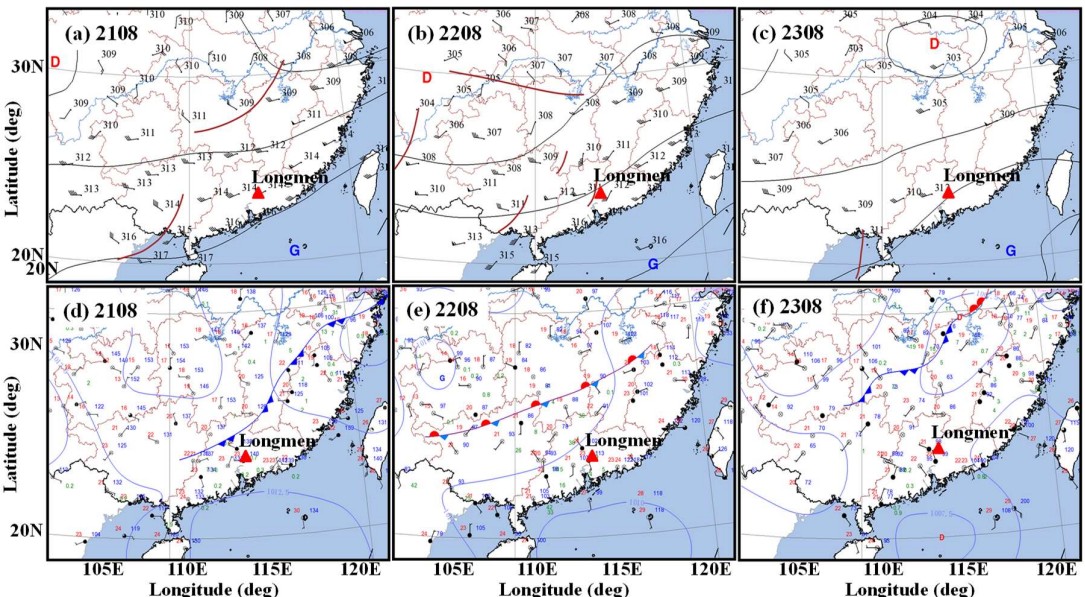

**Figure 2.** Weather charts on 21–23 April 2016: (**a**–**c**) at 700 hPa and (**d**–**f**) on the surface.

The combination reflectivity (CR) observed by Heyuan CINRAD, which is a Chinese network S-band weather radar located approximately 45 km in the east of the Longmen site, was further analyzed to briefly view the formation and evolution of cloud–precipitation on 21–23 April. Figure 3 shows nine moments of CR images of the cloud–precipitation that appeared near the Longmen site between 21–23 April. Several different kinds of convection were generated continually from the west and moved eastward under the guidance of the upper airflow. On 21 April, multi-cell convections affected the local site and moved quickly with apparent changes. Figure 3a–c illustrates that some new cells were formed continually and merged into the convection. Some old cells also dissipated gradually. On 22 April, a squall line system (Figure 3d) arrived over the Longmen station at 0836 BJT, with a bow echo exceeding 500 km. The local site was controlled by convective–stratiform mixed convections when the bow echo passed (Figure 3e–h). In the afternoon of 23 April, the radiosonde indicated that the atmosphere was in an extremely unstable condition, such that a large number of new cells were born in the local site. Figure 3i illustrates that new cells merged and transformed to become multi-cell convections.

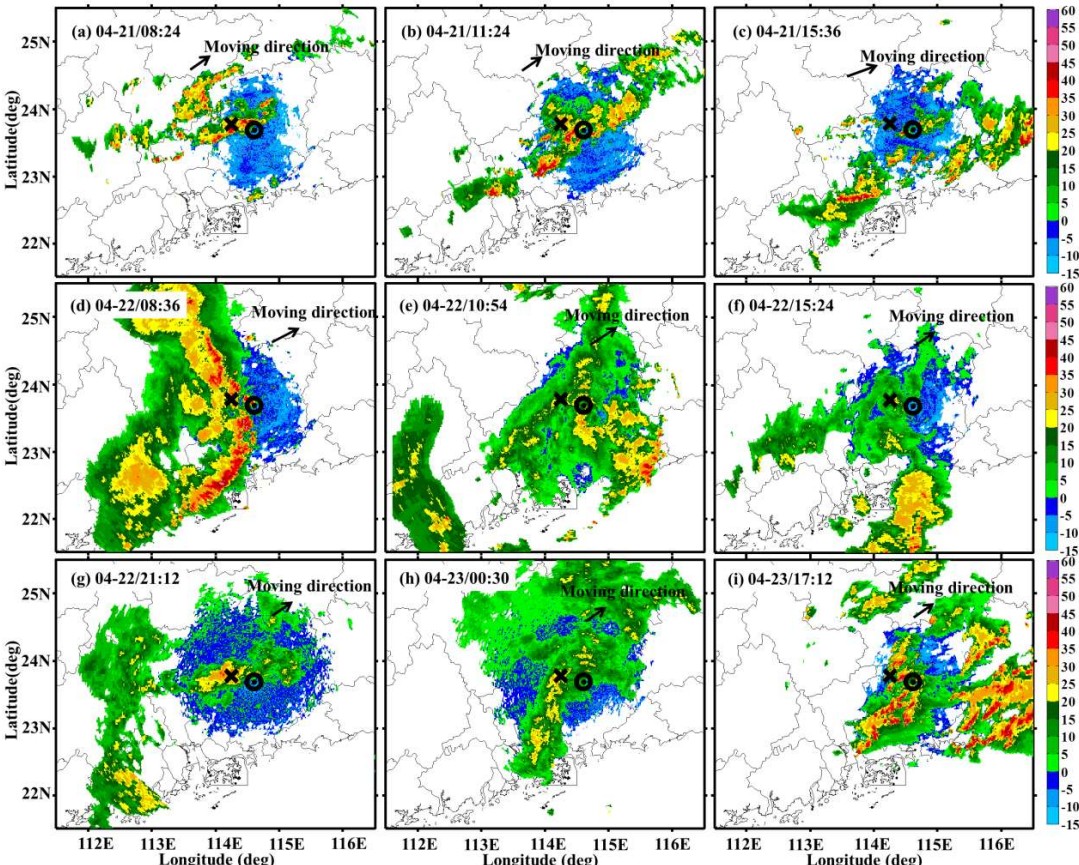

**Figure 3.** S-band radar combination reflectivity (CR) images of different convections between 21–23 April. (**a–i**) represent nine different moments. The circle represents the location of the Heyuan S-band radar, the cross represents that of the Longmen site.

### 3.2. Vertical Structures and Microphysical Properties of Different Convections

Figure 4 shows an overall time–height cross-section of the convective cloud–precipitation that occurred over the Longmen site on 21–23 April 2016. Comparing the reflectivity images obtained by the S-band radar and the MMCR, the reflectivity detected by the two radars coincided well, with a similar appearance and vertical structure in both time and space. As expected, although the S-band radar data were interpolated, the MMCR still detected much more sophisticated results than the S-band radar, because of its higher spatial and temporal resolution. Notably, the MMCR clearly showed a melting layer with a remarkable bright band at ~4 km, which was ambiguous in the S-band radar image. Meanwhile, the MMCR was able to detect weaker echoes on the cloud top and the boundary than the S-band radar, as a result of its high sensitivity. At the low level (i.e., less than 3 km), the MMCR was able to avoid the clear air turbulence echoes that appeared in the S-band radar image, because of its short wavelength. Figure 4 illustrates that the MMCR also had two deficiencies. First, the MMCR was affected by clutter under 2 km caused by plankton in the boundary layer, which has been reported by some researchers [10,57,58]. Second, the reflectivity of the MMCR was weaker than that of the S-band radar because of the signal attenuation and Mie scattering. By analyzing the deviation between the attenuation correction and the original, it was found that the underestimation could be compensated by 0–7 dB. The attenuation was proportional to the intensity of the reflectivity. In other words, the deviation was quite small, with a value under 1 dB for the clouds and weak precipitation at Z less than 20 dBZ. However, it became apparent, with a value exceeding 1 dB and reaching up to 7 dB, when the Z was greater than 20 dBZ. Using the relationship k = 0.28R [43,59], we also investigated the theoretical attenuation under different precipitation rates of this cloud–precipitation

process. The statistics indicate the ground R was mostly quite small, with 95.05 % of the values smaller than 5 mm h$^{-1}$, and caused a k less than 1.4 dB km$^{-1}$. A total 4.95% of R were able to reach higher than 5 mm h$^{-1}$ and cause more considerable attenuation.

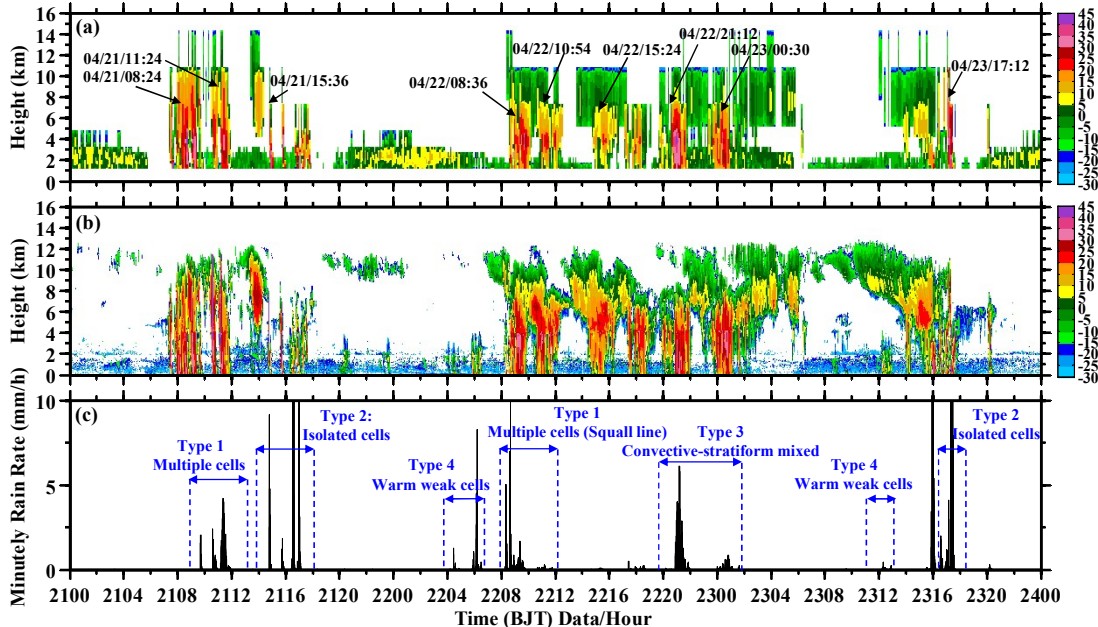

**Figure 4.** Overall time–height cross-sections of the cloud–precipitation that happened over the Longmen site on 21–23 April 2016: (**a**) S-band radar reflectivity, (**b**) MMCR reflectivity, and (**c**) rain rate measured by Parsivel on the ground.

To further analyze the vertical structures and microphysical properties of different convective cloud and precipitation, four types of convection (i.e., nonlinear and linear multi-cell, isolated-cell, convective–stratiform mixed, and warm-cell convections) were classified and picked out (Figure 4c). Individual analyses are presented in the subsections that follow.

### 3.2.1. Multi-Cell Convection

Multi-cell convection is a kind of mesoscale convective system with multiple and uniformly organized storm cells. It can exhibit both nonlinear and linear shapes, based on the weather background.

#### a. Nonlinear Multi-Cell Convection

Figure 5 exhibits the time–height cross-sections of the MMCR measurements, ground rain rate, and DSD for the nonlinear multi-cell convection that occurred over the site between 10:20–12:00 BJT on 21 April. Two periods of deep convective cloud passed through, with cloud top heights in the range of 10–12 km. The intense center, with a reflectivity exceeding 25 dBZ, hung at 5–9 km, indicating that the convection was still in the mature stage. The ice particles began to melt, with the presence of a bright band, as the height decreased to ~4.4 km. The bright band was closely related to the particle phase changes, and led to a sudden variation of Z, $\overline{V}$, $\sigma_v$, and LDR. Specifically, for the melting process, the water films attached on the ice crystal surface can strengthen the radar backscattering energy to generate increases in Z and LDR. The particles shrunk with a higher density as the ice crystals transformed into raindrops, causing faster falling speeds and a decrease of the radar-measured $\overline{V}$. In radar sampling volume, the scattered Doppler velocities of the particles caused by the phase change led to an increase of $\sigma_v$.

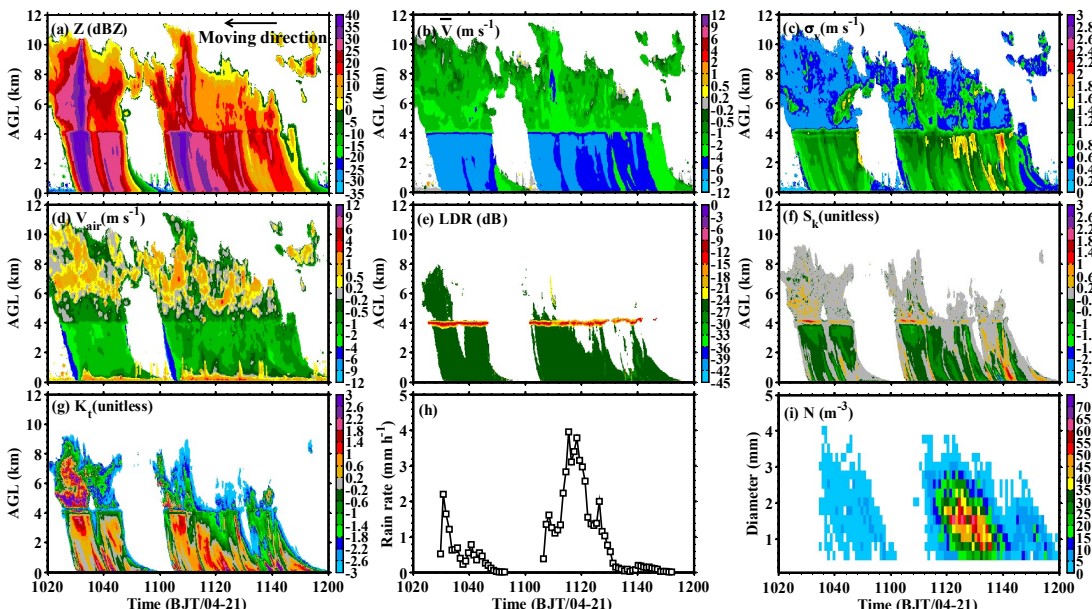

**Figure 5.** Time–height cross-sections of the MMCR measurements and Parsivel-measured rain rate and DSD for a nonlinear multi-cell convection that occurred over the site between 10:20–12:00 BJT on 21 April. (**a–g**) are radar reflectivity, mean Doppler velocity, spectrum width, vertical air velocity, linear depolarization ratio, spectral skewness, and spectral kurtosis; (**h,i**) are ground rain rate and drop size distribution.

Although the usage of MMCR's Z, $\overline{V}$, $\sigma_v$, and LDR for melting layer observation has been investigated in previous studies [60,61], radar-observed $V_{air}$, $S_k$, and $K_t$ for further inferring the dynamic and microphysical properties in cloud–precipitation interiors have not yet been widely used. Figure 5d–g shows the retrieved $V_{air}$, which indicates that the updrafts occurred mainly at the low level and the upper part of the cloud body. The updrafts had a medium value of less than 4 m·s$^{-1}$. Below the melting layer, the air motions turned to regular downward motions because of the sinking of cold air and hydrometeor evaporation. Above the melting layer, the $S_k$ and $K_t$ values were both positive in a particular area, with an intense reflectivity located between 5 and 9 km at 10:30 BJT. This observation illustrates that the ice crystal growth was more apparent in this area, located in front of the cloud moving direction and the corresponding ascending inflow. In the other areas, the $S_k$s were close to 0, while $K_t$s were negative, suggesting that the ice crystal growth was relatively slow in the cloud. The $S_k$ and $K_t$ values in the melting layer changed rapidly and reacted to the variations of the particle phase and size distribution.

Figure 5h,i shows the ground rain rate and the DSD measured by Parsivel. Compared to those in Figure 5d, the rain rate and the DSD in these figures were highly relevant to the updraft thickness. The rain rate peaks and large raindrops appeared in the moments wherein deep updrafts existed. Moreover, the slanted updraft not only led to a skewed cloud body, but also delay the ground rainfall for a few minutes.

The profiles of the averaged radar measurements between 10:20–12:00 BJT on 21 April are also presented herein. Figure 6a–h clearly shows the variations of Z, $\overline{V}$, $\sigma_v$, LDR, $V_{air}$, $S_k$, $K_t$, and Doppler spectra at different heights. We also further explain the detailed microphysics and dynamics in the cloud interior at different height ranges.

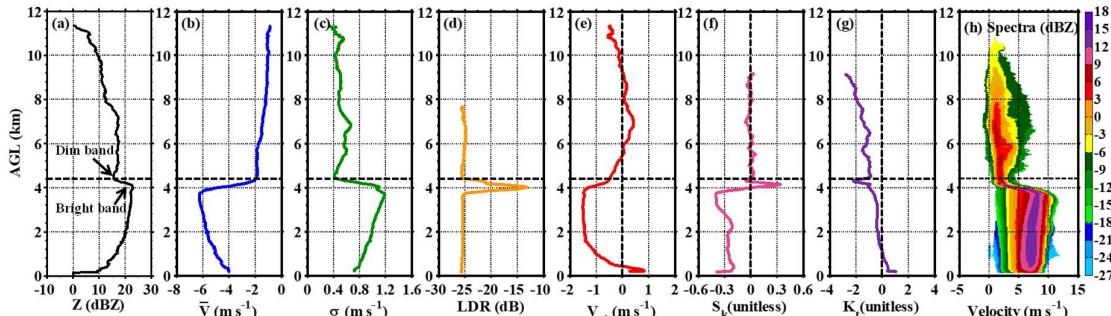

**Figure 6.** Profiles of the averaged radar measurements and mean Doppler spectra at different heights observed between 10:20–12:00 BJT on 21 April. (**a**–**h**) are reflectivity, mean Doppler velocity, spectrum width, linear depolarization ratio, vertical air velocity, spectral skewness, spectral kurtosis, and Doppler spectra.

The radar-measured Z near the cloud top (>9 km) increased gradually from 0 to 11 dBZ with a slight decline of $\overline{V}$, indicating that the growth of the ice particles in this range was dominated mainly by an increase of the particle number concentration. The ice particle size was small, and could not yield enough perpendicular backscattered energy to provide a valid LDR. The cloud signal in the radar Doppler spectra was also too weak or narrow to provide accurate $S_k$ and $K_t$ values.

The Z value in the upper part of the cloud from 9 to 5 km maintained an increase from 11 to 17.5 dBZ, with a decrease of $\overline{V}$ from −1 to −2 m·s$^{-1}$. The $\sigma_v$ exhibited the same variation with $V_{air}$, i.e., the larger $\sigma_v$ corresponded well to the stronger updrafts. The LDR maintained a low value near −25 dB; $S_k$ was close to 0; and $K_t$ became a smaller negative value. These change features of the radar measurements implied that some of the larger ice crystals were formed under the favorable influence of relatively strong updrafts. As shown in the Doppler spectra, the radar signals in this height range constituted both plenty of small ice particles and a few larger ice crystals with broad shapes.

A unique feature can be observed near the melting layer, that is, the Z profile showed a decline of approximately 3 dB near the middle of the cloud from 5 to 4.4 km. This interesting phenomenon is called a "dim band" [62], which can be triggered by large ice crystals with diameters greater than 2.48 mm for the Ka-band MMCR as a result of Mie scattering. As a response, radar $\overline{V}$ had no change, $\sigma_v$ became smaller, and the Doppler spectra apparently shrunk.

The radar measurements in the melting layer from 4.4 to 3.6 km had much more sensitive changes. First, in the upper part of the melting layer, small ice particles melted into droplets, while large ice crystals attached water films to their surface, leading to sudden increases in Z, $\sigma_v$, and LDR and decreases in $\overline{V}$ and $V_{air}$. $S_k$ varied from near 0 to a distinct positive value, while $K_t$ changed to a larger negative value. These characteristics indicate that the signal components of both droplets and large melting ice crystals changed in the radar Doppler spectra. At the center of the melting layer, Z and LDR approached maximum values of 22 dBZ and −13 dB, respectively. Notably, $S_k$ returned to 0, indicating that the scattered radar signal returned to be a normal distribution. Accordingly, the Doppler spectra were equivalently dominated by rain droplets and melting ice crystals. In the lower part of the melting layer, all ice crystals melted completely into raindrops, and some droplets began to collide and merge into larger raindrops, leading to decreases in $\overline{V}$, LDR, and $S_k$, and increases in $\sigma_v$ and $K_t$. Large raindrops mostly dominated the Doppler spectra.

Raindrop collision and breaking occurred simultaneously under the melting layer. The latter ruled mainly the radar measurements, with gradual decreases in Z and $\sigma_v$, and increases in $\overline{V}$ and $V_{air}$. The relevant left-shifted raindrop signals can be seen clearly in the Doppler spectra image.

As analyzed earlier, the microphysical and dynamic properties from the cloud top to the ground in the multi-cell convection experienced significant changes in terms of the hydrometeor phase, phase transformation, componental proportion, air motion variation, etc., and can be closely related to the variations of the MMCR measurements and the Doppler spectra. A forward pattern of the Ka-band

MMCR Doppler spectra for seven different height ranges in the multi-cell convection is presented in this section. Figure 7 illustrates that the shapes, amplitudes, widths, and velocity positions of the patterns can together represent the microphysical and dynamic changes in the interior of the cloud and the precipitation.

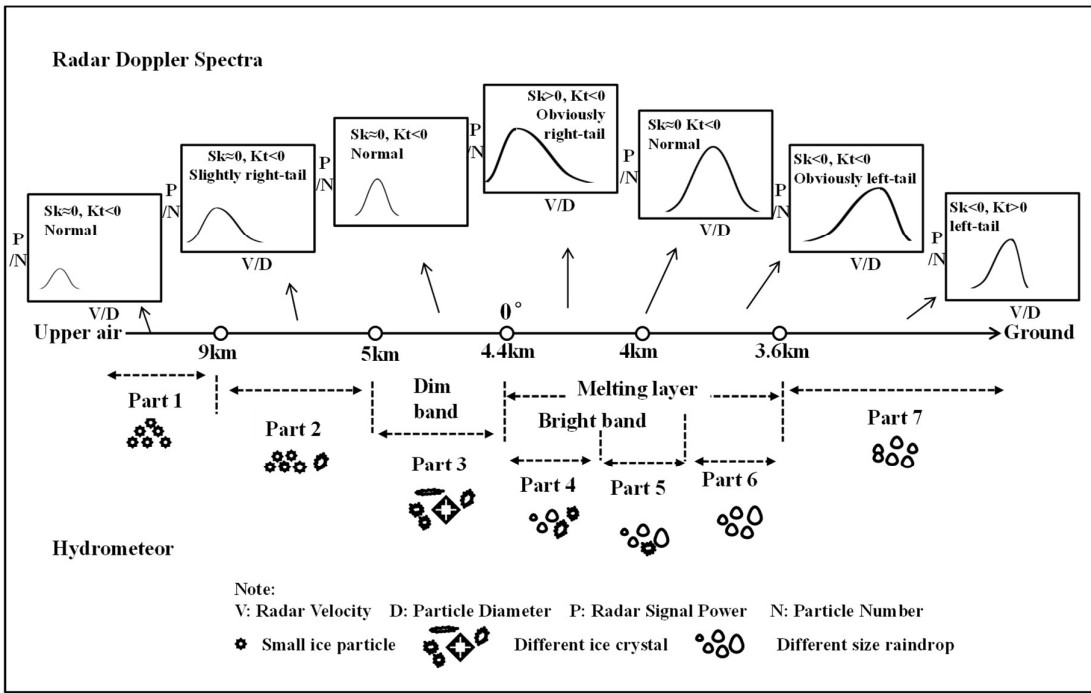

**Figure 7.** Forward pattern of the Ka-band MMCR Doppler spectra for seven different height ranges in the multi-cell convection.

b. Squall Line Convection

A squall line convection organized by a row of storm cells was observed by the MMCR from 08:00 to 12:20 BJT on 22 April. Figure 8 shows the time–height cross-sections of radar measurements and the time series of the Parsivel-measured rain rate and DSD. The cloud and precipitation of the squall line had a larger horizontal scale with a 4 h duration. The cloud top was relatively stable, with a height of approximately 9 km. From 08:00 to 09:00 BJT, some new cells with a columnar shape gradually formed in the downwind area in front of the convection (circled area), and merged into the system because of the strong updrafts. The radar-detected images indicate that the maximum Z of the new cell center exceeded 30 dBZ, with a maximum $V_{air}$ greater than 12 m·s$^{-1}$ and the largest $\sigma_v$ greater than 3 m·s$^{-1}$. This result suggests that the strong updraft in front of the squall line was slanted and rapidly brought low-level warm and wet airflow into the cloud body. Consequently, the hydrometeors yielded intensively and grew, and the cells produced strong rainfall on the ground. During this period, the ground rain rate and the DSD showed two showers, corresponding to two updrafts. The first shower had a low rain rate of approximately 5 mm·h$^{-1}$ and a maximum raindrop diameter greater than 3 mm. The relevant updraft was relatively shallow and weak. However, the second shower had a significant rain rate of approximately 23 mm·h$^{-1}$ and a maximum raindrop diameter over 4 mm. The corresponding updraft was stronger and thicker.

The MMCR-measured images from 09:00 to 10:00 BJT corresponded to the cells of the convection's linear-shaped part. The radar measurements showed similar vertical structures and features as the multi-cell in the last subsection. In the period from 10:00 to 12:30 BJT, the followed anvil of the linear-shaped cells was mixed, with newly generated convective clouds that developed further to form larger non-spherical ice crystals. As the rectangular areas on the images, the LDRs in this area exhibited relatively high values, while $S_k$ and $K_t$ were mostly positive, implying that the ice particles

had an evident growth process in the cloud interior. However, the ground rainfall was very small in this period, and revealed an existing strong evaporation process at the low level.

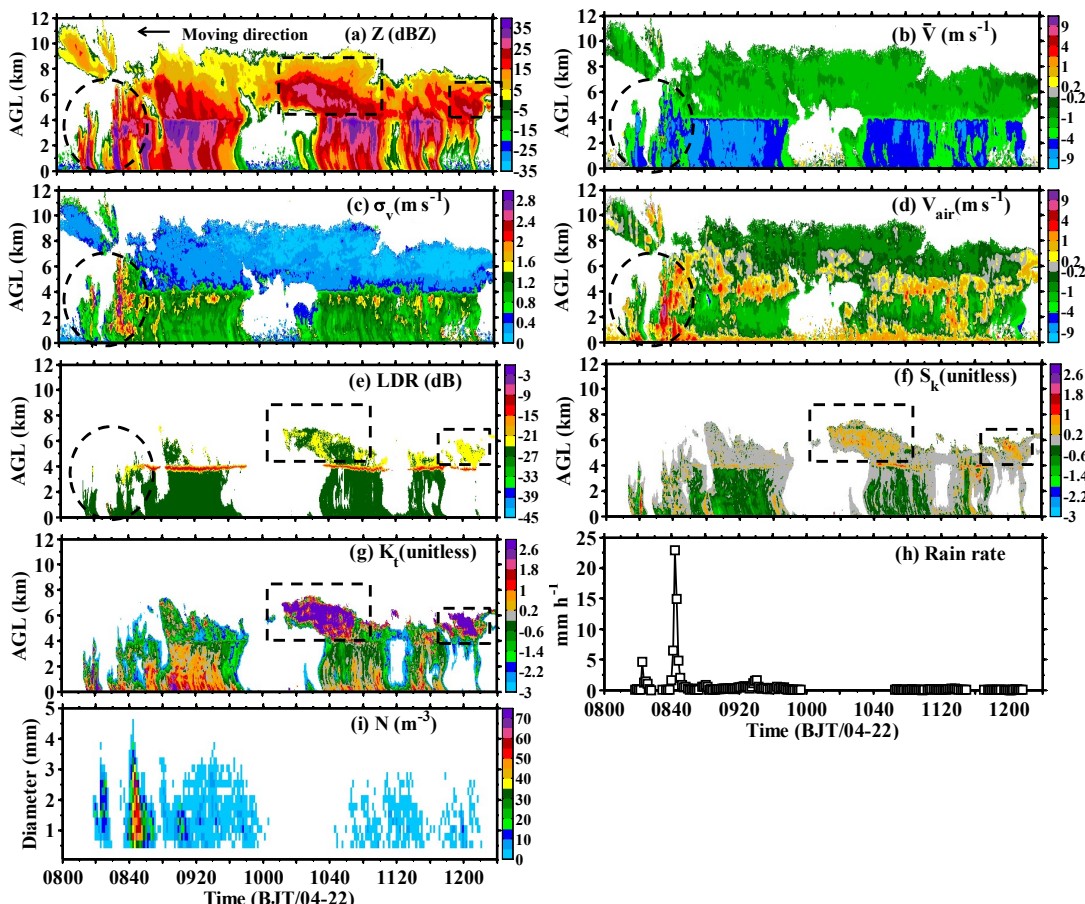

**Figure 8.** Time–height cross-sections of the MMCR measurements and the Parsivel-measured rain rate and DSD on the ground for a squall line convection that occurred over the site between 08:00–12:20 BJT on 22 April. (**a–g**) are radar reflectivity, mean Doppler velocity, spectrum width, vertical air velocity, linear depolarization ratio, spectral skewness, and spectral kurtosis; (**h,i**) are ground rain rate and drop size distribution.

### 3.2.2. Isolated-Cell Convection

Five isolated cells passed the Longmen site between 14:34–17:10 BJT on 21 April. Figure 9 shows the corresponding MMCR measurements, ground rain rate, and DSD. All five isolated cells had a columnar structure with an upward-convex cloud-top, the heights of which ranged from 6 to 8 km. The cells had a small horizontal scale, passing through within 20 min, and an intense $Z$, with a maximum over 35 dBZ. A comparison of the radar measurements with the ground rain rate and the DSDs of the five cells showed that they exhibited quite different vertical structures, which reflected different physical states in the cloud interior. Figure 9a depicts that Cell 1 was the most convective, with the strongest and deepest updrafts that could rush up to the cloud top. Note that no melting features existed on the radar images. The updraft speed was sometimes higher than 4 m·s$^{-1}$, but inhomogeneous, and the updraft variation near the cloud top caused apparent wind shear (Figure 9c) related to the observed large $\sigma_v$. Large hydrometeors yielded in the cloud, corresponding to large $\bar{V}$ values under 4 km, where the updrafts could no longer hold up the big hydrometeors. Although Cell 2 had the highest cloud top (close to 8 km), it began to weaken with the presentation of a bright band at 4.2 km. The related radar measurements showed the same characteristics as those mentioned in Section 3.2.1. For the other three cells, several new cumuli occurred gradually in front of the cells

because of the updrafts, which were slanted and staggered with the downdrafts in space to cause a small-scale alternative variation in the cloud. Compared to Cell 3, Cells 4 and 5 had similar vertical structures, but much smaller rain rates and raindrop sizes, because the hydrometeors in Cells 4 and 5 were almost hung in the air.

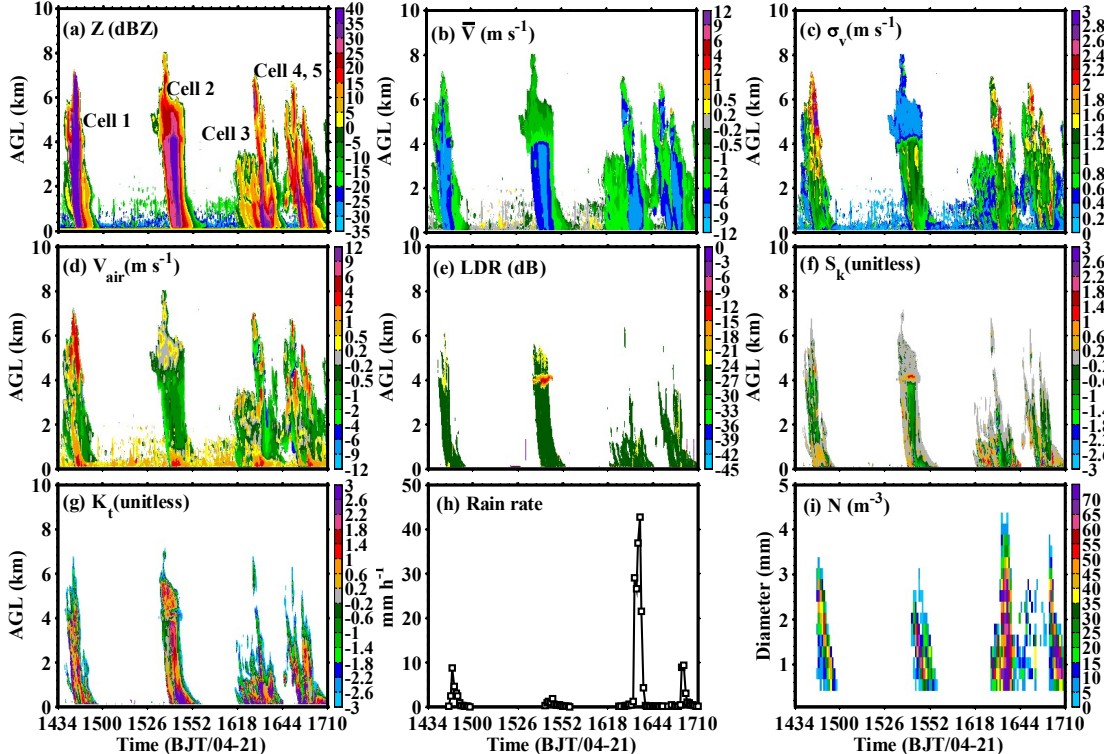

**Figure 9.** Time–height cross-sections of the MMCR measurements and the Parsivel-measured rain rate and DSD for a series of isolated cells between 14:34–17:10 BJT on 21 April. (**a**–**g**) are radar reflectivity, mean Doppler velocity, spectrum width, vertical air velocity, linear depolarization ratio, spectral skewness, and spectral kurtosis; (**h**,**i**) are ground rain rate and drop size distribution.

We further state herein the critical effects of the updrafts, with inter-comparison of the observations among the five cells. An updraft is generally correlated with the hydrometeor growth process and rainfall, that is, a strong updraft can result in a large rain rate and large raindrops [63,64]. The updraft slope is also an essential factor. If the updraft produced at the low level is skewed, it will not be easily washed by the downdraft, and can bring sufficient water vapor into the cloud body to supplement the system (e.g., Cells 3, 4, and 5). By contrast, if the updraft tends to be straight, it is restrained by an overlap with the downdrafts, such that the corresponding rain rate and raindrop size are also limited.

### 3.2.3. Convective–Stratiform Mixed Cloud–Precipitation

The MMCR observed a period of convective–stratiform mixed cloud–precipitation (CSMC) from 22 April, 20:00 BJT to 23 April, 02:00 BJT. Figure 10 shows the time–height cross-sections of the radar measurements and the time series of the ground rain rate and DSD. From 21:50 to 23:30 on 22 April, a layer of stratocumulus was observed over the head, with a height ranging from 6 to 9 km. The stratocumulus was relatively weak, with a maximum Z under 15 dBZ, and produced a small amount of precipitation, but quickly evaporated in the air. At other times, the clouds were mostly precipitating CSMCs, with apparent melting features of the radar measurements. The melting layer height was near 4.5 km. Two strong CSMCs observed from 20:45 to 21:50 BJT and 23:45 to 01:08 BJT both had weak updrafts at the low level and above the melting layer. The updrafts were suppressed by collision from the upper downdrafts. As measured by Parsivel, the former CSMC produced stronger

rainfall and more giant raindrops than the latter. One of the explanations for this observation is that the low-level updraft of the former was relatively stronger, and had a strong echo with a larger Z. Moreover, the hydrometeor of the former touched the ground, while that of the latter still hung in the air. Figure 11 also shows the mean profiles of these two CSMCs. Their vertical structures were similar to those of the multi-cell presented in Figure 7, with the only difference being that a dim band above the melting layer could be found in the CSMC profiles because the hydrometeor in the cloud–precipitation was not large enough to induce Mie scattering.

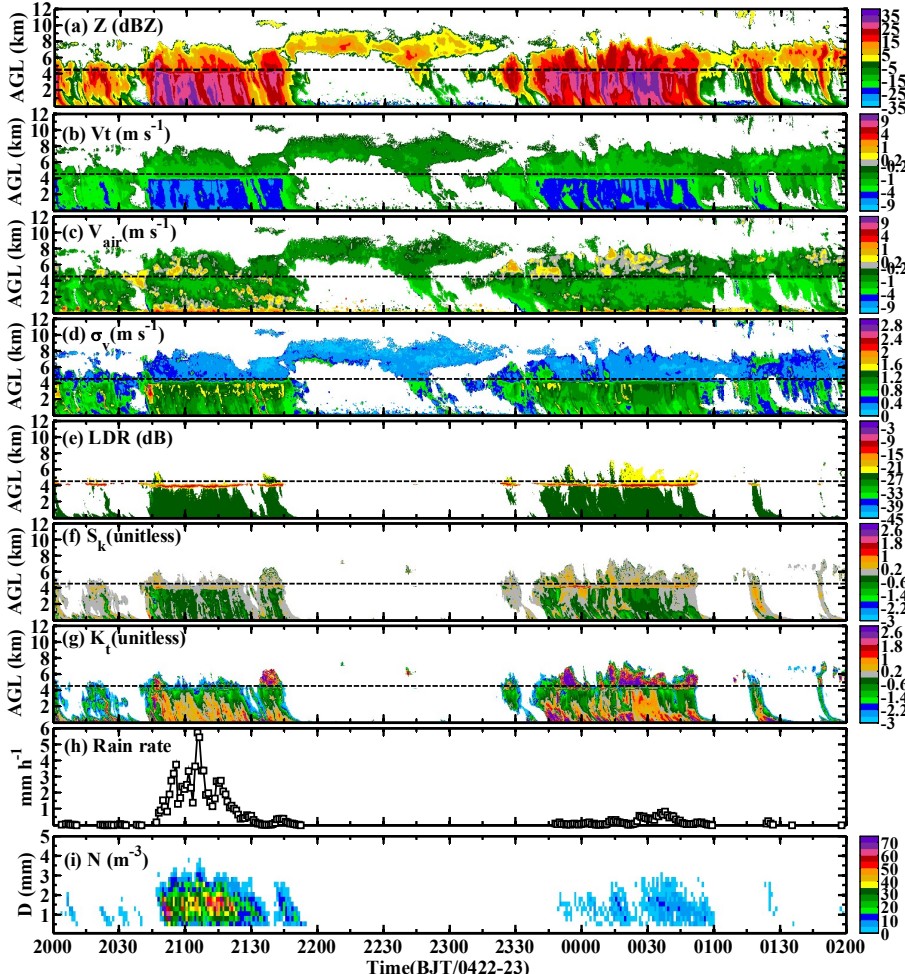

**Figure 10.** Time–height cross-sections of the MMCR measurements and the ground rain rate and DSD for a period of CSMCs obtained from 20:00 BJT on 22 April to 02:00 BJT on 23 April. (**a**–**i**) are radar reflectivity, particle terminal velocity, vertical air velocity, spectrum width, linear depolarization ratio, spectral skewness, and spectral kurtosis; (**h**,**i**) are ground rain rate and drop size distribution.

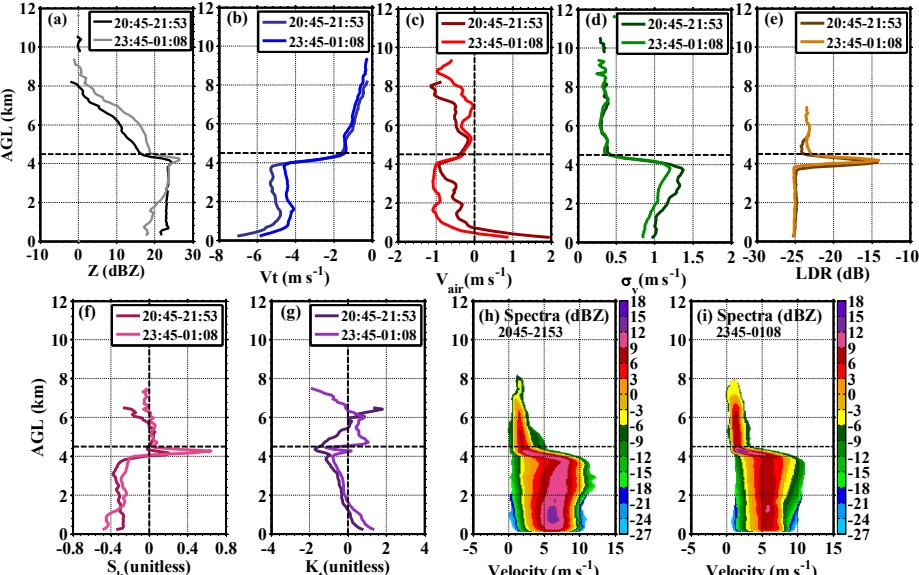

**Figure 11.** Profiles of the averaged radar measurements and mean Doppler spectra at different heights observed from 20:45 to 21:50 BJT on 22 April and 23:45 to 01:08 BJT on 23 April. (**a–i**) are reflectivity, particle terminal velocity, vertical air velocity, spectrum width, linear depolarization ratio, spectral skewness, spectral kurtosis, and Doppler spectra.

### 3.2.4. Warm Cells

Except for the deep, cold convections analyzed earlier, several warm cells with cloud tops under a 0° layer were also captured by the MMCR between 05:42–06:37 BJT on 22 April. The radar measurements presented in Figure 12 showed that the warm cells had different macroscopic features compared to the aforementioned cold cells. The warm cells had smaller horizontal scales, with a passing time of approximately 15 min and cloud top heights under 3 km. Their appearance exhibited a straight and columnar shape under the influence of updrafts. The updrafts occurred nearby and controlled the whole cloud body to bring plenty of water vapor from the low level and provide favorable dynamic conditions. The cold cells had complicated physical processes in the interiors of the clouds as a result of the hydrometeor phase change, while the warm cells had relatively simple physical processes and features. Moreover, the microphysical parameters in the warm cells could be easily derived because of the specific geometrical shape and the relationship between the falling speed and the diameter of the hydrometeor.

Figure 13 shows the retrieved microphysical parameters, including the LWC, rain rate (R), particle mean diameter ($D_m$), and total number concentration ($N_{total}$) for the warm cells. The microphysical properties for the warm cells could be characterized by comparing the retrievals with the radar measurements. First, the LWC and R had positive correlations with the Z and $V_t$ (mean falling velocity of particles) distributions, because these four quantities were proportional to both particle diameter and concentration, and the high LWC and R values corresponded to strong reflectivity and fast falling speeds, with maximums of LWC and R possibly approaching 0.35 g·m$^{-3}$ and 50 mm·h$^{-1}$ for the warm cells, respectively. However, this coherence was quite different on a boundary of 500 m.

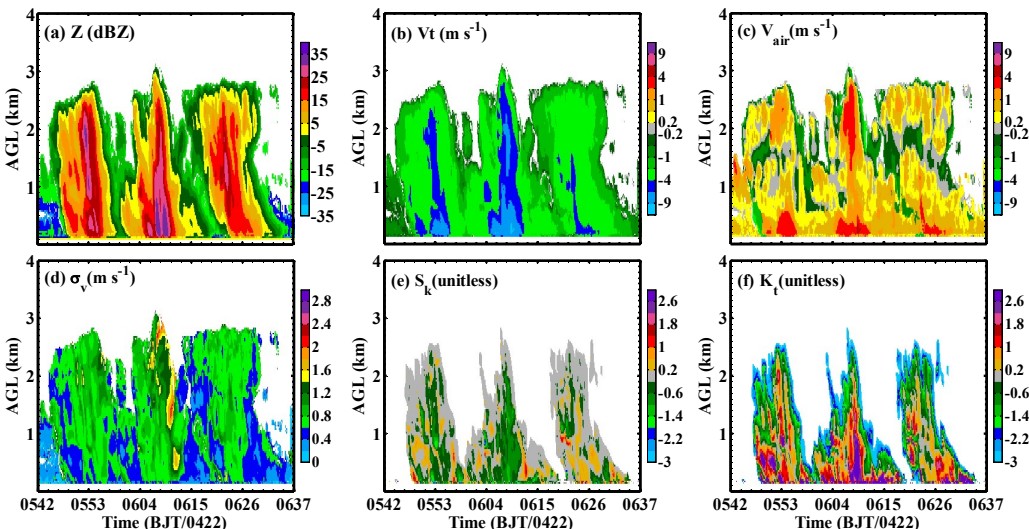

**Figure 12.** Time–height cross-sections of the MMCR measurements observed from 05:42 to 06:37 BJT on 22 April. (**a**–**f**) are reflectivity, particle terminal velocity, vertical air velocity, spectrum width, spectral skewness, and spectral kurtosis.

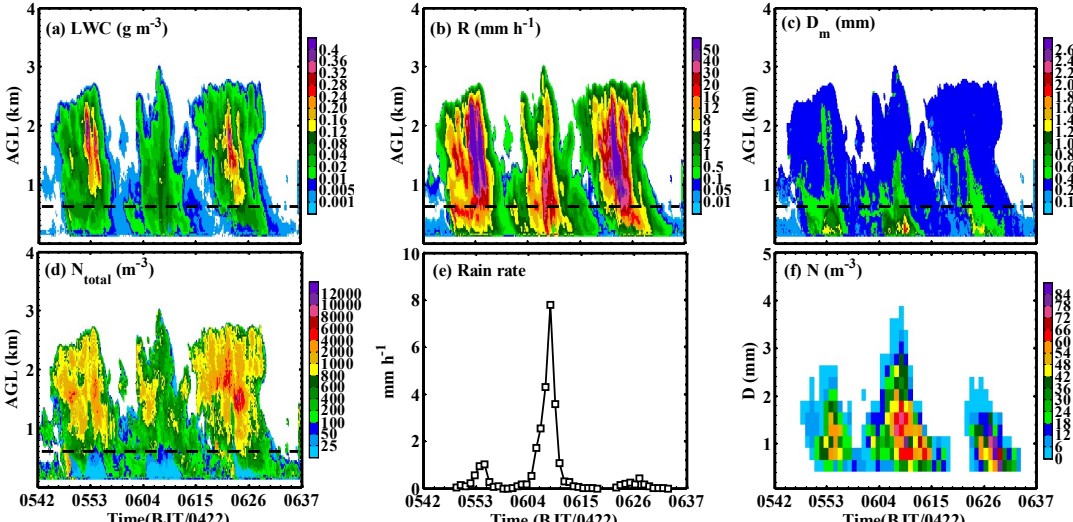

**Figure 13.** Time–height cross-sections of the MMCR-retrieved microphysical parameters and the ground rain rate and DSD observed from 05:42 to 06:37 BJT on 22 April. (**a**–**d**) are radar-derived liquid water content, rain rate, particle mean diameter, total number concentration; (**e**,**f**) are ground rain rate and drop size distribution.

The strong reflectivity area in the cloud in the upper height range (>500 m) had a large LWC and strong R, corresponding to relatively small $D_m$ and large $N_{total}$. In other words, the cloud body was occupied by both a larger number of small droplets and a few large raindrops, which had equivalent contributions to form a rather homogeneous particle spectrum. The $D_m$ and $N_{total}$ values ranged mainly from 0.2 to 0.6 mm and 400 to 8000 $m^{-3}$. The LWC and R in the lower height range (≤500 m) were much smaller than expected from Z and $V_t$, and had maximums of 0.08 $g·m^{-3}$ and 20 $mm·h^{-1}$, respectively. The $D_m$ and $N_{total}$ showed that the particles at the low level were much bigger, with diameters ranging from 0.5 to 2.2 mm, and the number concentration was small, ranging from 1 to 800 $m^{-3}$.

The inconsistent results in the two above-mentioned height ranges led to a further investigation, performed by comparing the Parsivel-measured ground rain rate and DSD with the radar measurements and retrievals. Figure 13e,f depicts that three peaks with large-sized raindrops and strong rain rates

presented on the ground. These peaks were related to the updraft intensity. The trends of the radar-retrieved R and $D_m$ were in a good agreement with the ground rain rate, and the DSD and their large values appeared coincidently in the time series. For the radar-retrieved $N_{total}$, it was basically coherent with the ground raindrop concentration at the upper height range. In contrast, it presented a negative correlation with the ground raindrop concentration at the lower height range. This discrepancy was caused by the effects of Mie scattering, signal attenuation, and signal supersaturation, which are discussed in Section 4.

## 4. Discussion

In this section, we further discuss the difference between the radar retrievals and the Parsivel observations. Accordingly, Z, R, $D_m$, and $N_{total}$ of the radar first available range gate at 150 m were compared with the measurements on the ground. Figure 14 shows that the trends of Z obtained by the two instruments were consistent. However, the Z values of the MMCR were smaller than Parsivel's, especially at the curve peaks, with a maximum bias of 10 dB. Three possible reasons for the weakening of the radar-measured Z can be cited. First, Figure 13f clearly shows that a specific part of the large raindrops with diameters greater than 2.48 mm, which can trigger Mie scattering for K-band MMCR, corresponded to the Z curve peaks. Thus, Mie scattering can occur and lead to a Z weaker than that in the real cloud–precipitation state. Second, the peaks of the MMCR Z curve were related to the large LWC, which was proportional to the radar signal attenuation. Therefore, the radar signal attenuation could be significant because of the large LWC. Lastly, the MMCR maximum detectable reflectivity at the low level was relatively small because of the limited receiver dynamic range. The maximum detectable reflectivity for the first available range gate was 18 dBZ [18], which means that the radar-measured Z was smaller than that expected because of the signal saturation. The R and $D_m$ curves of the two instruments also had the same variation trends, while the radar results were smaller than the ground observations for the same reasons as Z. The R biases on the three rainfall peaks were 2.0, 1.8, and 2.5 mm·h$^{-1}$, respectively. The $D_m$ biases were ~0, 0.27, and 0.2 mm, respectively.

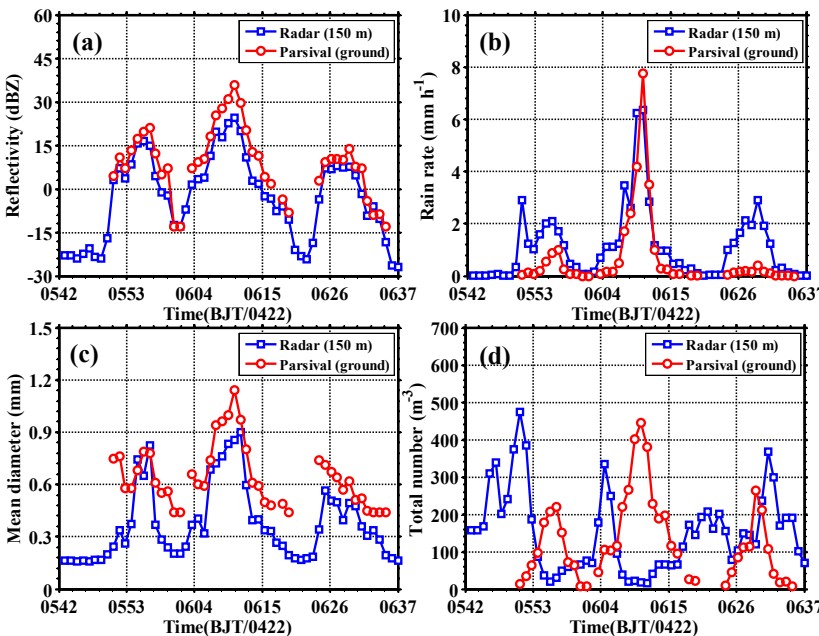

**Figure 14.** Time series of (**a**) reflectivity, (**b**) rain rate, (**c**) particle mean diameter, and (**d**) total number concentration measured by the MMCR (at the first available range gate, 150 m) and Parsivel (on the ground).

By contrast, the radar-derived $N_{total}$ revealed an inverse correlation with the ground measuring result. Moreover, this discrepancy existed mainly under 500 m, which is the height level shown as the

marked line in Figure 13. That is, above this height level, the radar's $N_{total}$ was basically coincident with the ground observations, whereas under this height level, the two instruments' results were opposite. The nearest radiosonde data were further investigated to explain the probable reasons for the observed phenomenon. The result showed that the lifting condensation level for the current atmosphere was 452 m. Thus, we can infer that the 500 m height level was close to the boundary of the cloud body and the fallstreak. Above this boundary, the radar targets involved small congealed cloud particles, while they consisted mainly of large-sized raindrops below the boundary. Under the same Z value, the small cloud particles could have a much larger $N_{total}$. In contrast, large raindrops have a smaller $N_{total}$, and the effects of signal attenuation, Mie scattering, and signal saturation can be serious. In this situation, the radar-derived results in the fallstreak suffered an apparent bias. The system bias of the two instruments could also have influenced the comparisons.

## 5. Conclusions

A vertical pointing Ka-band MMCR with high spatiotemporal resolution and high sensitivity can provide measurements including original Doppler spectra, reflectivity, mean Doppler velocity, spectrum width, linear depolarization rate, skewness, and kurtosis. In addition, with appropriate data processing and retrieval methods, many key microphysical and dynamic parameters of cloud–precipitation can be further obtained, including, but not limited to, the mean particle diameter, total number concentration, liquid water content, rain rate, air vertical velocity, and mean particle falling speed. These radar measurements and retrievals provide ample information on cloud properties, and can be valuable for further studies in atmospheric science, such as cloud–precipitation physics, climate change, weather modification, etc.

This study focused mainly on the vertical structures and microphysical features of different kinds of convective cloud–precipitation in South China during the pre-flood, season using a solid-state vertical pointing Ka-band MMCR and a laser disdrometer. The correlated data processing and retrieval procedures for the instruments were presented. Subsequently, four kinds of convections, namely, multi-cell, isolated-cell, convective–stratiform mixed, and warm-cell convections, were analyzed in detail. The results showed that the multi-cell and convective–stratiform mixed convections had similar vertical structures of their cloud–precipitation body. They experienced nearly the same microphysical processes in terms of particle phase change, particle size distribution, hydrometeor growth, and breaking. A forward pattern was proposed to specifically characterize the vertical structure, which provided the MMCR Doppler spectra models reflecting the different microphysical and dynamic features and variations in the different parts of the cloud body. Moreover, for the Ka-band radar, an apparent bright band was found in both multi-cell and convective–stratiform mixed convections. However, a dim band was found only in the multi-cell convection. The dim band was a reaction of the Mie scattering effect, which can be indirectly used to conclude the maximum size of the large hydrometeor in the interior of a cloud. The cloud–precipitations of the isolated-cell and warm-cell convections had relatively small horizontal scales and exhibited columnar structures with upward-convex cloud tops. Vertical air motion played key roles in cloud formation and development, and deeply affected the rainfall properties on the ground. Strong and slanted updrafts can bring sufficient water vapor into a cloud body and hold up the falling hydrometeor to cause stronger showers, which have a higher rain rate and larger raindrops. By contrast, weaker updrafts can correspond to weaker showers and related smaller rainfall on the ground.

Based on the relationship between liquid particle diameter and its falling speed, the LWC, R, $D_m$, and $N_{total}$ of the warm-cell convection were derived herein using radar Doppler spectra. The radar-retrieved results revealed that the upper part of the cloud body was dominated mostly by small droplets with a large concentration, while the lower part showed large raindrops with a small concentration to have a greater contribution as a result of the particle growth process. The depth and intensity of the updrafts had a significant positive correlation with the cloud microphysical properties and ground rainfall features.

Compared with the ground observations of the disdrometer, the radar retrievals at 150 m (a height of the first available radar range gate) were basically reliable. The results of the two instruments were coincidental for the LWC, R, and $D_m$. For the $N_{total}$, the results were also coherent with each other; however, under the cloud base, the radar result was quite different from the disdrometer. The reasons for this may include two aspects. The first is the radar signal weakening caused by serious attenuation, oversaturation, and Mie scattering. The second is the instrument sensitivity and height difference, that is, the MMCR with a high sensitivity has a certain ability to detect small droplets in the air, while the disdrometer can only measure raindrops with a diameter greater than 0.312 mm, leading to a large gap of $N_{total}$.

**Author Contributions:** Conceptualization, J.Z. and L.L.; methodology, J.Z.; software, J.Z., P.Z.; validation, J.Z. and Y.L.; formal analysis, J.Z., P.Z.; investigation, Y.L.; resources, L.L.; data curation, J.Z., P.Z. and Y.L.; writing—original draft preparation, J.Z.; writing—review and editing, Y.C., L.L.; visualization, Y.C.; supervision, P.Z.; project administration, J.Z.; funding acquisition, J.Z.

**Funding:** This research was funded by the National Natural Science Foundation of China (Grant Nos. 41705008, 41605022) and Science Project of Headquarters of State Grid Corporation of China.

**Acknowledgments:** The authors would like to thank the Chinese Academy of Meteorological Sciences for providing the radar data.

**Conflicts of Interest:** The authors declare no conflict of interest.

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
