# Peer review of "A Study of Vertical Structures and Microphysical Characteristics of Different Convective Cloud–Precipitation Types Using Ka-Band Millimeter Wave Radar Measurements"

_remotesensing, doi:10.3390/rs11151810_

Round 1

Reviewer 1 Report

Please find the reviews as attached.

Author Response

Please find the responses as attached.

Reviewer 2 Report

Paper provides a comparison between mm-wave cloud radar and S-band radar.  It is shown that the former provides higher resolution imagery than the latter even though interpolation is used in the latter to improve image resolution.  

Well-written and sound scientiifc methodology.  There are a few minor Engligh spelling and  grammatical errors so the paper should be read carefully thourgh to correct errors.  

Author Response

Please find the responses as attached.

Reviewer 3 Report

The paper is aimed to describe the application of MMCR measurements to the characterization of convective cloud precipitation microphysical and dynamical processes.

Besides this, the authors examine a very specific case of precipitation, and the analysis evidences the presence of four kinds of convections. The variety of convection processes and conditions helps to test the methodology results in different conditions.

In the present form, the manuscript looks adequate for publication as far as the presentation of results is concerned. Nevertheless, several aspects have to be improved before publication. Here some recommendations for the authors:

1) The biggest issue with the present form of the manuscript regards the description of the methods and the retrieval of the target quantities. Along with those, the authors should give values for the uncertainties that affect the retrieved quantities, and how this compare to previous results.

2) Lines 50-55: this passage of the introduction is way too specific compared to the rest of the section. I would rephrase, and perhaps put the numbers in the discussion of the instrument specifics (Section 2.1).

3) Lines 98-100: based also on the content of the paper, this looks like the main point of the work, and deserves a deeper discussion. I would recommend adding some references describing previous works on the same topic (e.g. https://journals.ametsoc.org/doi/full/10.1175/JAMC-D-13-0311.1 by Min Deng et al., and https://journals.ametsoc.org/doi/full/10.1175/MWR3321.1 by Stephens and Wood). Based on this, the authors should also clearly state what specific advancement(s) the present work brings to the scientific community, beyond the application to a specific case.

4) Line 176: the acronym LDR is not defined anywhere in the paper. Please define it.

5) Line 207: (mW): the watt unit is indicated with capital letter W and has to be corrected.

6) Line 221-222: it is stated that the data from the first two channels of Parsivel were discarded. Are this the first two channels describing the smallest or the largest particles? Based on the text, I would believe the first, but it has to be explicitly stated.

Author Response

Please find the responses as attached.

Round 2

Reviewer 3 Report

I believe the authors have satisfactorily addressed all my concerns and comments. I have no further comments on this manuscript, which I recommend to be published.

Author Response

Dear reviewer,

Thank you so much for your kind work. We really appreciate your valuable comments and 

recommendations. 

Your sincerely,

All authors, Jiafeng Zheng, Peiwen Zhang, Liping Liu, Yanxia Liu, Yuzhang Che